# Tuberculosis screening, isoniazid preventive therapy coverage and factors associated with active TB diagnosis among people living with HIV at public health facilities of central Ethiopia

**Mecha Aboma***, **Bayisa Abdisa, Gudata Imana, Kefyalew Taye, Gonfa Moti, Merob Fufa**

College of Health Science and Referral Hospital Ambo University, Ambo, Ethiopia

* abomamecha@gmail.com, mecha.aboma@ambou.edu.et

## Abstract

### Introduction

TB is the most frequent opportunistic infection and cause of death among People living with HIV/AIDS. Human immunodeficiency virus-positive individuals are routinely screened for tuberculosis as a means of monitoring efforts to mitigate the consequences of the disease on HIV-positive patients. TB status assessment identifies HIV-positive clients who show no evidence of active TB by symptom screening and would benefit from treatment with isoniazid for prevention of TB disease among HIV positives.

### Methods

A facility-based cross-sectional study design was conducted in public health facilities of West Shewa Zone from January to February 2019. Of the 28 Public health facilities providing HIV/AIDS care and support (HCT and ART services) 13 of them were selected by simple random sampling techniques. Finally, 815 study participants were recruited by systematics sampling techniques & proportional sample size allocation was applied depending on the HIV patient load in each health facility. After reviewing relevant literature, a structured questionnaire adapted from standardized WHO guidelines, prepared for monitoring & evaluation of TB/HIV activities was used to collect data via interviewer.

### Results

Of the total, 769 (94.4%) PLHIV were screened for tuberculosis, among which 212 (27.6%) were found to be positive for active tuberculosis. Among 557 (72.4%) individuals eligible for IPT, only 300 (53.9%) were provided IPT; 257 (46.1%) eligible PLHIV were not provided IPT. Resident (adjusted odds ratio [AOR] 5.6), those who didn't attend school (AOR 4.0), primary school (grade 1-8) (AOR 3.2), and secondary school (grade 9-12) (AOR 4.2) were significantly associated with the likelihood of tuberculosis infection.

**Data availability statement:** The dataset used and analyzed throughout this study is uploaded as Supplementary Information.

**Funding:** The author(s) received no specific funding for this work.

**Competing interests:** The authors have declared that no competing interests exist.

## Conclusion

The present study findings demonstrated that tuberculosis screening for PLHIV at West Shewa Zone public health facilities was improved in comparison with reports from many African countries and other parts of Ethiopia. The IPT implementation rate fell short of both national and WHO guidelines, notwithstanding this improvement. In the study area, diagnostic methods for tuberculosis and existing preventive measures should be improved overall.

## Introduction

Mycobacterium tuberculosis is the pathogen bacterium responsible for tuberculosis (TB) development. In 2022, tuberculosis was the second leading cause of death from a single infectious agent, following COVID-19. and was the main killer of individuals living with HIV [1,2]. Among people living with HIV, the risk of developing tuberculosis is 20–37 times higher [3]. Approximately 10.6 million new TB cases were reported globally, reflecting an incidence rate of 133 cases per 100,000 population, with 6.3% of individuals living with HIV. Tuberculosis deaths totaled 1.3 million, including an estimated 167,000 among HIV-positive individuals, predominantly occurring in the African Region (68%). The TB mortality rate among HIV-positive individuals in Africa was 9.5 per 100,000. In 2022, the WHO noted 25.6 million people living with HIV in Africa, with 167,000 deaths attributed to TB-HIV co-infection. The region accounted for 73% of global TB-HIV cases and 81% of deaths in 2019 [2,4].

Ethiopia is among the 30 countries with the highest rates of both TB and TB/HIV co-infection in the world with the overall mortality-weighted prevalence of TB-HIV co-infection was 15.27%. The WHO Global TB Report 2021 estimated a TB incidence rate of 132 cases per 100,000 population and a TB incidence rate of 8.6 per 100,000 among HIV-positive individuals in Ethiopia [5]. The WHO and the Ethiopian Federal Ministry of Health have jointly developed and suggested three specific actions related to TB and HIV within the framework of essential services for preventive measures, medical care, and therapies for HIV and tuberculosis (TB). Implementing these activities incorporates the delivery of TB and HIV services by applying "the three I's," which include intensive case finding for tuberculosis, administering isoniazid preventive therapy, and establishing infection prevention measures within healthcare facilities [6,7]. Isoniazid preventive therapy (IPT) serves as a crucial public health strategy aimed at reducing the incidence of tuberculosis in people living with HIV [8]. Despite the issuance of this commendation, merely 2.3 million people have undergone screening for tuberculosis, with only 178,000 being provided with isoniazid preventive therapy [9].

In 2007, Ethiopia started implementing IPT; however, due to several obstacles, including stigma, accessibility to medical facilities, adverse drug reactions, and a lack of IPT training, social support, supportive supervision, good client-provider communication, standard operating procedures, quality adherence counseling, and patient registers, the program's implementation has remained very low [10–12].

To avert the development of a clinical tuberculosis illness, Isoniazid preventive therapy is provided to people with HIV who have a recurrent tuberculosis infection [13].

The most crucial step that comes before deciding to start IPT is screening to rule out active TB in patients who are HIV positive. In adolescents and adults with HIV infection, the regimen for TB preventive therapy is isoniazid 300 mg pyridoxine 50 mg (vitamin B6) daily for at least 6 months, as recommended by the WHO [14,15]. For HIV-positive patients who were confirmed negative for TB, the administration of IPT is deemed effective since it significantly

reduces the likelihood of TB, with a risk reduction ranging from 33% to 67% throughout one to nineteen years [15–17].

Despite significant advancements in targeting individuals affected by TB, the success of initiatives aimed at reducing the incidence of TB among those infected with HIV remains considerably below the goals outlined in the global strategy to eliminate TB [9]. There is inadequate data on the extent to which progress has been made in implementing IPT among eligible people living with HIV, particularly in Ethiopia. Thus, in this study, we aimed to assess TB screening coverage and IPT among PLHIV in the west Shewa zone of Oromia Regional State, Ethiopia.

## Methods

### Study area

We conducted a facility-based cross-sectional study among PLHIV & following their HIV/AIDS care and support from January to February 2019 at public health institutions in the West Shewa zone, Oromia regional states. Ambo town is the capital city of the west Shewa zone and is located 115km away from Addis Ababa, the capital of Ethiopia [18].

### Study design, sample size, and sampling procedure

In this study, we included randomly selected adult PLHIV who were following their HIV/AIDS care and support at public health facilities and available during the data collection period. We determined sample size using a single population proportion formula taking 39% of the proportion of active TB among PLHIV [19] and a 95% confidence interval, 5% margin of error, and 10% nonresponse rate, and considering the design effect, the total sample size calculated and used in this study was 815.

Hence out of 28 public health facilities providing HIV/AIDS care and support (HCT and ART services) in zone 13 were selected by simple random sampling techniques and proportional sample size allocation depending on the HIV patient load in each health facility. The HIV registration books determined the total number of patients in each healthcare facility. Finally using systematic random sampling, samples were selected from the logbooks of PLHIV (Fig 1).

### Data collection tool

A structured questionnaire was used to collect data via the interviewer administer approach. After reviewing relevant literature, the tool was adapted from standardized WHO guidelines, and prepared for monitoring & evaluation of TB/HIV activities [3]. The interview was conducted in a private room following the provision of their HIV/AIDS care and support services. Full informed consent was obtained from all eligible participants after explaining the study's objectives to them. After obtaining informed consent, data collectors administered a questionnaire to respondents in their language. In addition to the interview, patient records were reviewed immediately after the interview to increase the validity of information and eliminate patient recall bias. Information was collected on the client's experience with the TB/HIV service provision & their status. Data collection was conducted by Nurses with close supervision by the principal investigator.

### Data management and analysis

The data analysis was done using SPSS version 21.0 software and we used descriptive analysis, such as frequency and percentage, to describe the socio-demographic characteristics of

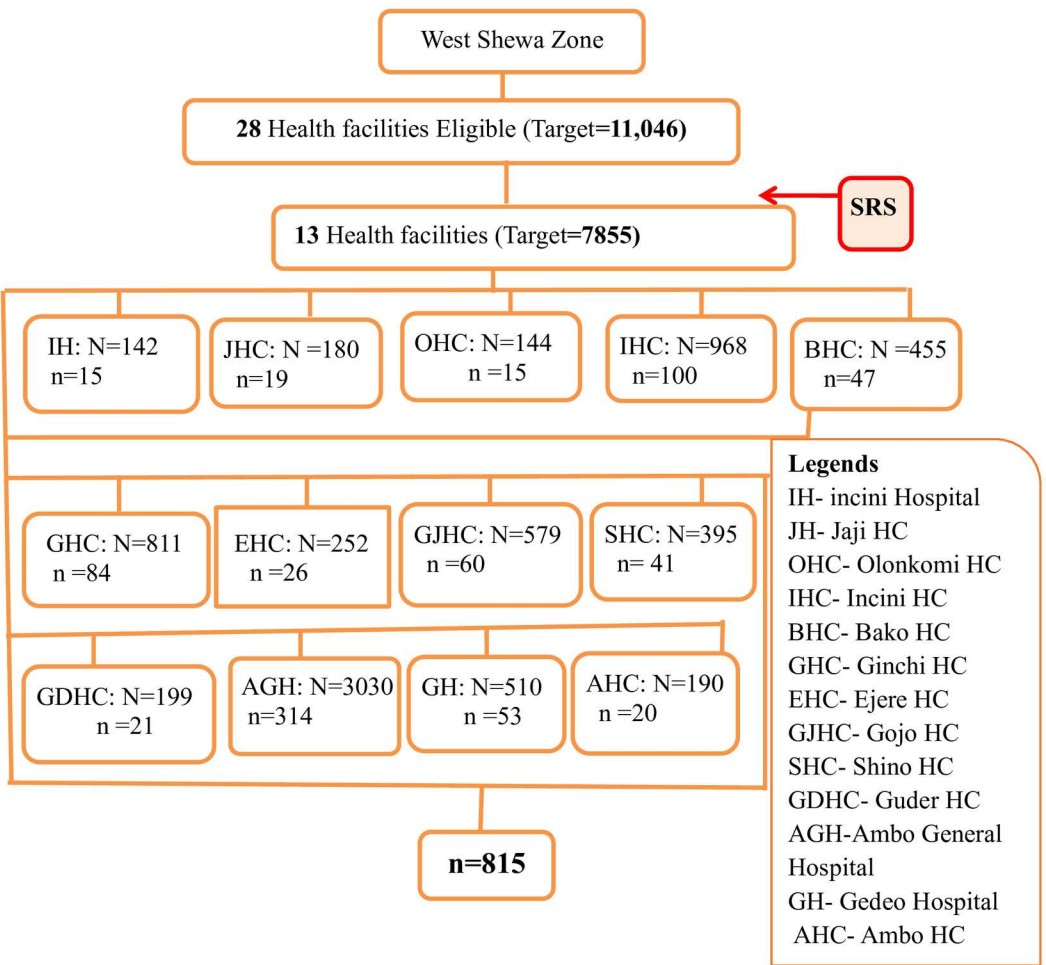

**Fig 1. Schematic presentation of sampling technique of public health facilities providing HIV/AIDS care and support (HCT and ART services) selected by Simple random sampling in districts of West Shewa Zone, Oromia, Ethiopia, 2019.**

participants. Binary logistic regression was employed to conduct both bivariate and multivariate analyses to determine the factors linked to tuberculosis infection among people living with HIV. Candidate variables in the final model (multivariate binary logistic regression) were identified using a binary logistic regression model with P < 0.25; in the final model, multiple logistic regressions were used to determine the independent effect of each explanatory variable on the study variable, with significance set to P < 0.05. The Institutional Review Board of Ambo University College of Health Science and Referral Hospital granted ethical approval.

## Algorithm for TB screening among adults and adolescents PLHIV

The national ART guidelines clearly state that all patients attending ART centers should be actively screened for TB. Patients with HIV who have untreated or undetected tuberculosis can visit ART facilities for care. Therefore, active efforts toward intensified TB case finding at ART centers are critical for early detection and treatment of TB. A clinical algorithm should be used to routinely and actively screen adults and adolescents living with HIV/AIDS for tuberculosis (TB). People who do not exhibit any of the symptoms, such as a current cough,

fever, weight loss, or night sweats, are not likely to diagnosis positive tuberculosis and should be considered for isoniazid preventive therapy (IPT). Individuals who report any one of the above symptoms are suspected of having active TB and should be assessed for TB and other co-infections.

But before starting IPT, the patient should be evaluated for any contraindications; after that, IPT is given right away. IPT is collected every month by the individual for daily use. Every visit should involve a side effect check and counseling regarding adherence for each patient. According to Ethiopian national guidelines, a tuberculosis skin test (TST) is unnecessary to initiate IPT in PLHIV. People living with HIV who receive a positive result from the Tuberculin Skin Test (TST) are likely to gain greater advantages from Isoniazid Preventive Therapy (IPT); therefore, TST can be utilized, when possible, to identify these individuals (Fig 2) [20].

Acid-fast bacillus (AFB) tests and chest X-rays are common diagnostic tools used to detect active TB among PLHIV at the studied health facilities. If PLHIV reports any symptoms that point to an ongoing TB lung infection, regardless of how long they have been present during a visit to the hospital or other healthcare setting, AFB testing is advised [18].

## Terms and operational definitions

**The Three I's.** To reduce the burden of TB disease among people living with HIV There are three activities, known as the "Three I's," that those providing care to people with HIV should do to protect them from active TB, help prevent active disease from developing, and to identify active TB disease early and improve the chances of cure:

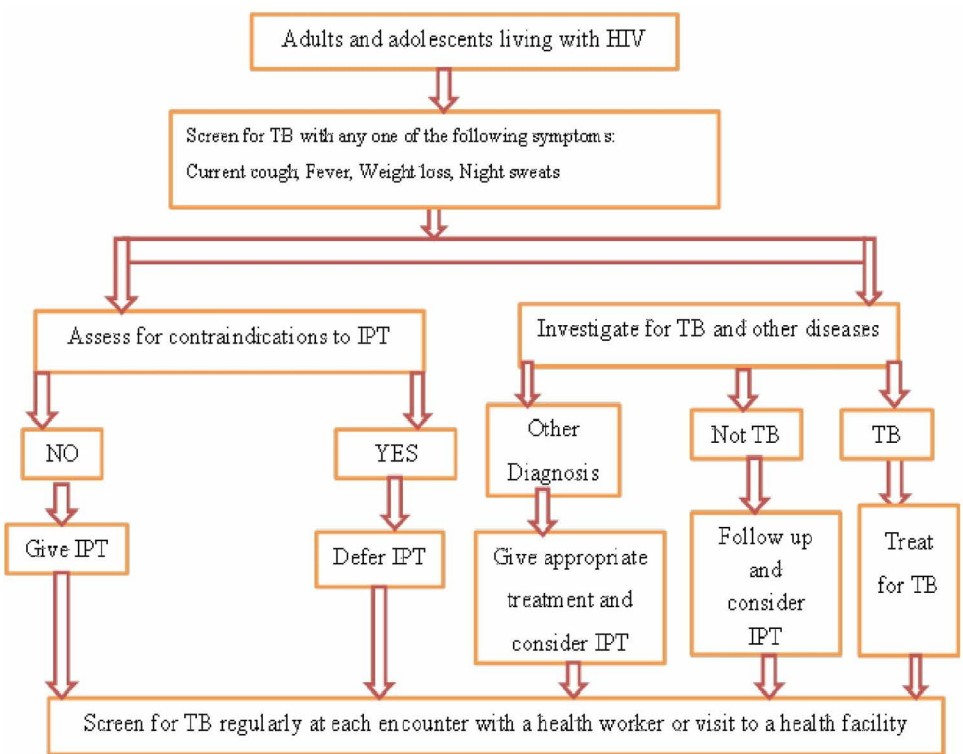

**Fig 2. The algorithm for TB screening in adult and adolescent HIV-positive settings with limited resources and a high prevalence of HIV [ 20].**

**ICF.** Intensified Case Finding for TB means regularly screening all people with or at high risk of HIV or in congregate settings who have active TB.

**IPT.** Isoniazid Preventive Therapy for TB can safely be given to people living with HIV without TB disease, reducing the risk of developing TB.

## Public and patient involvement

This study does not include any patients or the public. Patients were not asked for feedback on the study's design, nor were they involved in creating outcomes that were pertinent to them or in interpreting the findings. The dissemination plan was developed without the patients' input.

## Ethical clearance and participation consent

Ethical clearance was obtained from the Institutional Review Board of Ambo University College of Health Sciences and Referral Hospital, and an official letter of cooperation was sent to each health facility included in the study. Written and signed informed consent was obtained from the study subjects after explaining the objectives and procedures of the study and their right to participate or to withdraw at any time during the interview. The Research and Ethical Review Committee also approved its ethical issues as no procedure affected the study subject and the data was used only for research purposes. For this purpose, a one-page consent form was attached to each questionnaire's cover page, stating the study's general purpose and confidentiality issues, which, before the interview, the data collectors discussed. Lastly, we attest that the Declaration of Helsinki was followed during this investigation.

## Results

### Characteristics related to the socio-demographic profile of individuals living with HIV

A total of 815 study participants participated in this study making a response rate of 100%. The mean age of the respondents was 39.22 ( ± 10.51 SD) years, and almost more than half of the respondents, 541 (66.4%) were in the age group of 30-49 years. The majority of the study participants, 719 (88.2%) were Oromo by ethnicity, 444 (54.5%) of them were Orthodox Christian followers, and 420 (51.5%) of them were rural residents. Of the 815 PLHIV, 421 (51.7%) were male and 394 (48.2%) were female patients; 455 (55.8%) patients were married and 149 (18.3%) were divorced.

Most respondents, 314 (38.5%) have attended primary education and 251 (30.8%) had no formal education. Among the remaining PLHIV, 150 (18.4%) had completed secondary school (grades 9–12) and 100 (12.3%) were diploma and above. Regarding the study participants' jobs, 198 people (24.3%) were farmers, 164 people (20.1%) were without employment, 152 people (18.7%) were domestic workers, 135 people (16.6%) had employment, 112 people (13.7%) were merchants, and 54 people (6.6%) were students (Table 1).

### The magnitude of TB screening and IPT among people living with HIV

Of the total, 769 (94.4%) PLHIV were provided TB screening services, among which 212 (27.6%) reported a positive diagnosis for active tuberculosis from the time they began receiving HIV/AIDS care and supplementary assistance; to receive TB treatment, each of these individuals was registered in the TB clinic. Among 557 (72.4%) patients with HIV who were eligible for IPT (screened for tuberculosis and found to be negative) only 300 (53.9%) were provided IPT; 257 (46.1%) patients who were eligible for IPT were not

**Table 1. The socio-demographics of people living with HIV at Public Health Facilities of Central Ethiopia, 2019.**

| Variables (n = 815) | Categories | Frequency | Present |
|---|---|---|---|
| Sex of Client | Male | 421 | 51.7% |
| | Female | 394 | 48.3% |
| Marital status | Married | 455 | 55.8% |
| | Single | 118 | 14.5% |
| | Divorced | 149 | 18.3% |
| | Widowed | 93 | 11.4% |
| Age | 18–29 | 135 | 16.6% |
| | 30–39 | 269 | 33.0% |
| | 40–49 | 272 | 33.4% |
| | >=50 | 139 | 17.1% |
| Educational Status | Illiterate | 251 | 30.8% |
| | Primary school (1–8) | 314 | 38.5% |
| | Secondary school (9–12) | 150 | 18.4% |
| | High school diploma & above | 100 | 12.3% |
| Occupational status | Employed | 135 | 16.6 |
| | Un-employed | 164 | 20.1 |
| | Merchants | 112 | 13.7 |
| | Farmer | 198 | 24.3% |
| | Student | 54 | 6.6% |
| | Housewife | 152 | 18.7% |
| Ethnicity | Oromo | 719 | 88.2% |
| | Amhara | 64 | 7.9% |
| | Others | 32 | 3.9% |
| Religion | Orthodox | 444 | 54.5% |
| | Protestant | 299 | 36.7% |
| | Wakefata | 34 | 4.2% |
| | Muslim | 32 | 3.9% |
| | Catholic | 6 | 0.7% |
| | Total | 815 | 100.0 |
| Residences | Urban | 395 | 48.5% |
| | Rural | 420 | 51.5% |
| | Total | 815 | 100% |

provided IPT for unknown reasons. And more than half of the eligible patients for IPT, 316 (56.7%) were not knowledgeable about eligibility for IPT. Of those patients provided with IPT, 300 (100%) were counseled on IPT utilization, 300 (100%) known from where they collect INH, and the majority, 211(70.3%) were knowledgeable about the duration of IPT utilization. Regarding TB disease classification, 212(100%) of the patients were Pulmonary positive, 185 (87.3%) were new TB disease category, and nearly half 111 (52.4%) completed treatment.

Majority of the patients, 753 (92.4%) were not admitted at least once for HIV/AIDS, 598(77.8%) were screened for TB after diagnosed for HIV, 204 (96.2%) were diagnosed by AFB (microscopy), 267 (32.8) were > 500 CD4 counts during data collection, 183 (22.5%) were > 500 CD4 counts at HIV diagnosis, and 323(39.6%) were 200 - 350 CD4 counts at ART initiation (Table 2).

**Table 2. Magnitude of TB screening and IPT among people living with HIV in Central Ethiopia, 2019.**

| Variables | Categories | Frequency | Present |
|---|---|---|---|
| Screened for TB | Yes | 769 | 94.4% |
| | No | 46 | 5.6% |
| | Total | 815 | 100% |
| Active TB diagnosis | Yes (+ve) | 212 | 27.6% |
| | No (-ve) | 557 | 72.4% |
| | Total | 769 | 100% |
| IPT Prophylaxis given | Yes | 300 | 53.9% |
| | No | 257 | 46.1% |
| | Total | 557 | 100% |
| Method of TB Diagnosis | AFB (microscopy) | 204 | 96.2% |
| | Chest x-ray | 8 | 3.8% |
| | Total | 212 | 100% |
| Time of screened for TB | Before being diagnosed with HIV | 171 | 22.2% |
| | After being diagnosed with HIV | 598 | 77.8% |
| | Total | 769 | 100% |
| Admitted to hospital at least once for HIV/ADS | Yes | 62 | 7.6% |
| | No | 753 | 92.4% |
| | Total | 815 | 100% |
| Knowledgeable about eligibility for IPT | Yes | 241 | 43.3% |
| | No | 316 | 56.7% |
| | Total | 557 | 100% |
| Duration of IPT | 1–2 months | 65 | 21.7% |
| | 3–4 months | 24 | 8.0% |
| | >= 5 months | 211 | 70.3% |
| | Total | 300 | 100% |
| From where INH was collected | HIV care clinic | 300 | 100% |
| Counselled on IPT | Yes | 300 | 100% |
| | Total | 300 | 100% |
| Current CD4 counts | < 200 | 93 | 11.4% |
| | 200–350 | 251 | 30.8% |
| | 351–500 | 204 | 25.0% |
| | > 500 | 267 | 32.8% |
| | Total | 815 | 100% |
| CD4 counts at HIV diagnosis | > 500 | 183 | 22.5% |
| | 200-500 | 367 | 45.0% |
| | <200 | 265 | 32.5% |
| | Total | 815 | 100% |
| CD4 counts at ART initiation | >350 | 253 | 31.0% |
| | 200-350 | 323 | 39.6% |
| | < 200 | 239 | 29.3% |
| | Total | 815 | 100% |
| TB disease Classification | Pulmonary positive | 212 | 100% |
| TB disease Category | New | 185 | 87.3% |
| | Relapse | 27 | 12.7% |
| | **Total** | 212 | 100% |

### Active TB-related factors among people living with HIV

After adjusting for relevant confounding variables, the outcomes of the multivariate logistic regression model revealed that only residency and education had an independently significant relationship with tuberculosis infection.

Hence, rural resident clients were 5.9 times more likely to develop M. tuberculosis infection than their urban counterparts (AOR 5.6; 95% CI 1.73-19.14; P < 0.05), and PLHIV those who didn't attend school, primary school (grade 1-8), and secondary school(grade 9-12) were 4.0, 3.2 and 4.2 times more likely to have active TB, respectively, than those diploma and above (AOR 4.0; 95% CI 1.84-8.6; AOR 3.2: 95% CI: 1.5-6.8; AOR 4.2: 95% CI: 2.0-9.4; P < 0.05) (Table 3).

## Discussion

In this study, we found that 94.4% of PLHIV had been screened by healthcare providers for the usual signs and symptoms of TB, using a clinical algorithm. Our research revealed that people living with HIV received more tuberculosis screenings than those studied in Harar by Geleto et al. The number stated that 75.2% of people living with HIV were screened for tuberculosis [21].

Our findings were also somewhat higher than those in Addis Ababa reported by Denegetu and Dolamo (92.8%) and by Wesen and Mitike in which 87.9% of people with HIV/AIDS were offered TB screening [22,23]. In a national survey conducted in Ethiopia, 71% of people with HIV/AIDS were offered TB screening, which is also lower than our results [24]. Similarly, a

**Table 3. Multivariable logistic regression analysis for predicting the risk of tuberculosis infections among HIV-positive people in Central Ethiopia, 2019.**

| Variable (n = 769) | Categories | COR (95% CI) | AOR (95% CI) | P-Value for (AOR) |
|---|---|---|---|---|
| Sex | Male | 1.3 (0.93–1.75) | 1.0(0.7–1.5) | 0.220 |
|  | Female | 1.00[+] | 1.00[+] |  |
| Age | 18–29 | 2.84 (1.6–5.1) [*] | 2.8 (0.95–5.32) | 0.142 |
|  | 30–39 | 1.4 (0.91–2.235) | 1.3 (0.79–2.1) | 0.282 |
|  | 40–49 | 1.3 (0.84–2.04) | 1.35 (0.85–2.1) | 0.179 |
|  | >=50 | 1.00[+] | 1.00[+] |  |
| Marital Status | Married | 1.03 (0.75–1.4) | 1.3 (0.84–1.91) | 0.515 |
|  | Others[*] | 1.00[+] | 1.00[+] |  |
| Education | Illiterate | 2.2 (1.3–362) * | 4.0 (1.84–8.6) * | 0.000 |
|  | Elementary (1–8) | 1.8 (1.13–3.0) * | 3.2 (1.5–6.8) * | 0.001 |
|  | Secondary (9–12) | 2.21 (1.3–3.94) * | 4.2 (1.9–9.4) * | 0.001 |
|  | Diploma & above | 1.00 [+] | 1.00[+] |  |
| Residence | Rural | 2.0(1.4–2.8) * | 5.6 (1.73–19.14) * | 0.000 |
|  | Urban | 1.00[+] | 1.00[+] |  |
| Occupational status | Employed | 0.51 (0.33–0.85) * | 1.3 (0.62–2.63) | 0.589 |
|  | Unemployed | 0.4 (0.26–0.62) * | 0.4 (0.22–0.6) | 0.121 |
|  | Farmer | 0.5 (0.31–0.71) * | 0.82 (0.50–1.33) | 0.331 |
|  | Others[**] | 1.00[+] | 1.00[+] |  |

[+]The group of references.

*P < 0.05. Confidence interval (CI), crude odds ratio (COR), adjusted odds ratio (AOR),

*others (single divorced, widowed),

**others (merchant, students, house wife).

study on TB/HIV integration services in Sub-Saharan Africa indicated that only 64% of newly registered people with HIV were offered TB screening [14].

The fact that a sizable percentage of PLHIV had TB screening is optimistic about the results of this study. This increment over time might be attributed to the improvement in HIV/TB treatment and care services at the study health facilities. The West Sheswa zone is also the largest zone of the Oromia regional state, in which health facilities are expanding rapidly, with an increased number of trained healthcare providers to meet the needs of high-risk populations. Our study's findings were somewhat inferior to those of a study conducted in northern Ethiopia, where 98.2% of people living with HIV received tuberculosis screening [25] in addition to the findings of another study in Ethiopia where 96% of patients with HIV enrolled in HIV care were screened for TB [26]. The findings of our study were also lower than those conducted in southwest Ethiopia, which found that 99.7% of PLHIV had been screened by healthcare providers for the usual signs and symptoms of TB, using a clinical algorithm. [27].

Our study findings showed that 212 (27.6%) patients with HIV infection were diagnosed with active TB during their follow-up HIV care. In this study, we found a lower proportion of active TB among HIV-positive people than those from studies conducted in other parts of Ethiopia and in some developed countries, in Ethiopia (39.0%), in Harar (29.8%), in Hong Kong (39%), and Pakistan (30.2%) [19,21,28,29]. The findings of our study showed that, compared to other regions of Ethiopia and several affluent nations, a substantially larger percentage of PLHIV received an active TB diagnosis throughout their HIV treatment and follow-up care, southwest Ethiopia, 8.3%, Addis Ababa in 2008 and 2011 (15.6% and 10.4%, respectively), Georgia (22%), in northeast Ethiopia (24.3%) [23–25,27,30]. These differences might be attributed to false negatives reported for AFB testing and chest X-rays in diagnostic services. Additionally, inconsistent active TB diagnostics might result from inadequate diagnostic equipment, poor microscopy quality, untrained healthcare providers, substandard laboratory practices, and insufficient quality control measures, particularly in resource-limited settings.

The research carried out in Addis Ababa found a modest proportion of co-infection: 32 (7%) HIV-positive individuals also had positive testing for active tuberculosis [31]. The detection rates in our study [32] are higher than those in a German study of 11,693 people infected with HIV, where 233 (2%) of the participants were found to have active tuberculosis, 62 at enrollment, and 171 during subsequent follow-up visits [32].

We found that among 557 clients eligible for IPT, only 300 (53.9%) were offered IPT during follow-up HIV/AIDS care. This is lower than the rate reported in southwest Ethiopia and Harar; where 66.5% and 78.7% of PLWHI were offered IPT during follow-up HIV/AIDS care respectively [21,27]. However, we found that our findings were greater than those of two previous studies conducted in Ethiopia, where 39% of eligible people had started IPT, and in Addis Ababa, where 28.7% of people living with HIV were administered Isoniazid preventive therapy [19,23].

In line with the global report on TB, the statistical data show that the IPT coverage is 15.1% in Ethiopia, 5.4% in Bangladesh, 8.0% in Myanmar, and only 3.0% in Nigeria [33]. We found that a significant percentage of eligible people living with HIV had started IPT throughout HIV subsequent follow-up, in contrast to the majority of findings from Ethiopia and other nations.

The current high rates of IPT among research participants may point to the subsequent improvements in TB/HIV collaboration services; however, these findings could also be explained by variations in study design, sample size, and study environment between studies.

In our study, education and residence showed a statistically significant association with the likelihood of active TB among PLHIV. Education significantly affected active TB, consistent

with other study findings from southern Ethiopia [34]. A possible explanation might be that educated people have better health service-seeking behavior, frequent visits to health facilities for services, and are engaged more in disease prevention and health promotion. Our study findings indicated that PLHIV those who lived in rural residences were at higher risk for active TB than those who lived in urban residences. This might be owing to people living in urban residents having more access to health information, health care services, and other infrastructure that enables them to fight TB.

## Conclusion

The present study demonstrated that the implementation of TB screening among individuals with HIV infection in the west Shoewa zone was higher than that in many African countries, as well as in other parts of Ethiopia. Additionally, education and residence were identified as factors that determine the likelihood of active TB among PLHIV. Nevertheless, despite advancements, IPT implementation fell short of national and WHO guidelines.

In general, putting into practice the current national and World Health Organization recommendations could improve tuberculosis diagnosis techniques and accessible prevention strategies. Ongoing support of healthcare workers through continuous capacity building and experience sharing can serve to advance the implementation of TB screening and IPT initiation.

Additionally, service providers must be part of the solution by providing targeted patient education and counseling (to rural residents along with less schooled people living with HIV) to raise knowledge of the advantages and significance of IPT and enhance its acceptance. Additionally, collaborative research comprising several regions of the country including health facilities and healthcare professional-related factors is recommended to provide a more balanced view of active TB, IPT provision, and potential risk factors among patients with HIV infection.

## Supporting information

**S1 File. TBHIV original raw data.**
(SAV)

## Acknowledgments

We express our gratitude to Ambo University College of Health Sciences and Referral Hospital, as well as the data collectors, study participants, and any other individuals who played a formal or informal role in the successful completion of this research.

## Author contributions

**Conceptualization:** Mecha Aboma.

**Data curation:** Mecha Aboma, Bayisa Abdisa, Gudata Imana, Kefyalew Taye, Gonfa Moti.

**Formal analysis:** Mecha Aboma.

**Funding acquisition:** Mecha Aboma.

**Investigation:** Mecha Aboma, Merob Fufa.

**Methodology:** Mecha Aboma, Merob Fufa.

**Project administration:** Mecha Aboma, Gudata Imana, Merob Fufa.

**Resources:** Mecha Aboma.

**Software:** Mecha Aboma, Merob Fufa.

**Supervision:** Mecha Aboma, Bayisa Abdisa, Gudata Imana, Kefyalew Taye, Gonfa Moti, Merob Fufa.

**Validation:** Mecha Aboma, Bayisa Abdisa, Gudata Imana, Kefyalew Taye, Gonfa Moti, Merob Fufa.

**Visualization:** Mecha Aboma, Bayisa Abdisa, Gudata Imana, Kefyalew Taye, Gonfa Moti, Merob Fufa.

**Writing – original draft:** Mecha Aboma, Merob Fufa.

**Writing – review & editing:** Mecha Aboma, Bayisa Abdisa, Gudata Imana, Kefyalew Taye, Gonfa Moti, Merob Fufa.

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
