## [Decision Letter · Decision Letter 0]

4 Dec 2024

PONE-D-24-31425Tuberculosis Screening, Isoniazid Preventive Therapy and Associated Factors Among People Living with HIV at Public Health Facilities of Central EthiopiaPLOS ONE

Dear Dr. Aboma,

Thank you for submitting your manuscript to PLOS ONE. After careful consideration, we feel that it has merit but does not fully meet PLOS ONE’s publication criteria as it currently stands. Therefore, we invite you to submit a revised version of the manuscript that addresses the points raised during the review process.

We look forward to receiving your revised manuscript.

Kind regards,

Rebecca F. Baggaley

Academic Editor

PLOS ONE

2. In the online submission form, you indicated that [The dataset used and analyzed throughout this study is available from the corresponding author upon reasonable request.]. All PLOS journals now require all data underlying the findings described in their manuscript to be freely available to other researchers, either 1. In a public repository, 2. Within the manuscript itself, or 3. Uploaded as supplementary information. This policy applies to all data except where public deposition would breach compliance with the protocol approved by your research ethics board. If your data cannot be made publicly available for ethical or legal reasons (e.g., public availability would compromise patient privacy), please explain your reasons on resubmission and your exemption request will be escalated for approval.

Additional Editor Comments (if provided):

Reviewers' comments:

Reviewer's Responses to Questions

**Comments to the Author**

1. Is the manuscript technically sound, and do the data support the conclusions?

Reviewer #1: Partly

Reviewer #2: Partly

2. Has the statistical analysis been performed appropriately and rigorously? 

Reviewer #1: Yes

Reviewer #2: Yes

3. Have the authors made all data underlying the findings in their manuscript fully available?

Reviewer #1: Yes

Reviewer #2: Yes

4. Is the manuscript presented in an intelligible fashion and written in standard English?

Reviewer #1: No

Reviewer #2: Yes

5. Review Comments to the Author

Reviewer #1: I would like to commend the authors for conducting this important research on a topic of great relevance for the continued reduction of TB cases and deaths. This study provides valuable insights into the TPT cascade and contributes to existing knowledge on this critical area of TB prevention.

However, I have a few key concerns that need to be addressed:

Major Concerns:

Research Structure: The study appears to have two distinct components: the first focuses on patient progression through the early stages of the TPT cascade (screening for active TB, determining TPT eligibility, and providing TPT), while the second examines factors associated with active TB. The title does not fully reflect the second part of the study, and it seems somewhat disconnected from the first. It might be more coherent to analyze factors related to TB screening and TPT provision rather than factors associated with active TB.

Terminology: Some terms are used incorrectly, making the paper harder to understand. For instance, "TB infection" is used instead of "active TB," or "recurrent tuberculosis infection". Improving the language throughout the paper will enhance clarity.

Conclusions Not Supported by Results: Several conclusions in the manuscript are not adequately supported by the data presented (lines 288-289, 296, 299-300). Please revisit these sections to ensure that the conclusions are based on the results.

Minor Revisions:

Introduction: The introduction could be shortened, and it would be helpful to update the data, using statistics from 2021 and 2022 instead of 2018 and 2020.

Aim of the Study: The phrase "progress in reducing the risk of TB" (line 88) feels inappropriate as a research aim. A more accurate aim would be "progress in implementing TPT."

Sample Size Calculation: The concept of "design effect" used in sample size calculation (line 101) is unclear. Please provide further explanation or clarification.

Figure 1: It would be more informative if the authors could display the overall number of PLHIV (People Living with HIV) in each facility and then provide the sample size for each. All the abbreviations should be added in the legend.

Patient Evaluation: It would be helpful to include information on how many patients had a positive symptom screening but did not complete further evaluation, as well as how many patients were found to have contraindications for TPT.

Diagnostic Methods: It is unclear why 551 patients were diagnosed with AFB smear while the initial screening was based on symptoms. Were all patients who were initially screened positive referred for AFB testing?

Typographical Error: There is a typographical error in the numbers; "796" is used instead of "769" in some calculations.

Table 3: I suggest restructuring Table 3. The second and third columns could be removed, and a column for p-values should be added to enhance the clarity of statistical significance.

I hope these suggestions help improve the quality and clarity of the manuscript. Best of luck with your revisions.

Reviewer #2: 1. The conclusion must mention the associated factors according to the title (‘Associated Factors Among People Living with HIV at Public Health Facilities of Central Ethiopia’).

2. This study still uses the AFB microscopy test to determine active TB. Is there no molecular rapid test available?

3. In data table 2, the CD4 variable (Current CD4 Count) should be added to group the CD4 count based on the diagnosis status of active or latent TB.

4. There were 257 (46.1%) eligible PLHIV patients who were not given IPT, please provide clear reasons for this.

6. PLOS authors have the option to publish the peer review history of their article (what does this mean? ). If published, this will include your full peer review and any attached files.

**Do you want your identity to be public for this peer review?** For information about this choice, including consent withdrawal, please see our Privacy Policy .

Reviewer #1: No

Reviewer #2: **Yes: ** Tutik Kusmiati

---

## [Author Response · Author response to Decision Letter 1]

30 Dec 2024

Author's response to Reviewers Comments

Manuscript ID: - PONE-D-24-31425

Title of Manuscript: Tuberculosis Screening, Isoniazid Preventive Therapy, and Associated Factors Among People Living with HIV at Public Health Facilities of Central Ethiopia

Authors:

1. Mecha Aboma: abomamecha@gmail.com

2. Bayisa Abdisa

3. Gudata Imana

4. Kefalew Taye

5. Gonfa Moti

6. Merob Fufa

A Point-by-point Response to Reviewer Comments to the Authors

Reviewer #1: Major Concerns:

1. Reviewer 1 Question 1. Research Structure: The study appears to have two distinct components: the first focuses on patient progression through the early stages of the TPT cascade (screening for active TB, determining TPT eligibility, and providing TPT), while the second examines factors associated with active TB. The title does not fully reflect the second part of the study, and it seems somewhat disconnected from the first. It might be more coherent to analyze factors related to TB screening and TPT provision rather than factors associated with active TB.

Authors Response 1. Yes, the comment is accepted and revised and indicated by highlighted with track change as flow: Tuberculosis Screening and Isoniazid Preventive Therapy Coverage and Factors Associated with Active TB Diagnosis Among People Living with HIV at Public Health Facilities of Central Ethiopia

2. Reviewer 1 Question 2. Terminology: Some terms are used incorrectly, making the paper harder to understand. For instance, "TB infection" is used instead of "active TB," or "recurrent tuberculosis infection". Improving the language throughout the paper will enhance clarity.

Authors Response 2. Yes, the comment is accepted and corrected and indicated by highlighting with track change as flow: TB infection" is replaced by "active TB," or "recurrent tuberculosis infection"

3. Reviewer 1 Question 3. Conclusions Not Supported by Results: Several conclusions in the manuscript are not adequately supported by the data presented (lines 288-289, 296, 299-300). Please revisit these sections to ensure that the conclusions are based on the results.

Authors Response 3: Yes, the comment is accepted and revised and indicated by highlighted with track change lines 306-3010 and 3017-3025

Minor Revisions:

4. Reviewer 1 Question 4. Introduction: The introduction could be shortened, and it would be helpful to update the data, using statistics from 2021 and 2022 instead of 2018 and 2020.

Authors Response 4. Yes, the comment is accepted and revised and indicated by highlighted with track change, Lines 41-65, 69-70, 73-74, and 76-79

5. Reviewer 1 Question 5. Aim of the Study: The phrase "progress in reducing the risk of TB" (line 88) feels inappropriate as a research aim. A more accurate aim would be "progress in implementing TPT."

Authors Response 5. Yes, the comment is accepted and revised and indicated by highlighted with track change as flows: There is inadequate data on the extent to which progress has been made in implementing IPT among eligible people living with HIV, particularly in Ethiopia. Line 100-104

6. Reviewer 1 Question 6. Sample Size Calculation: The "design effect" concept used in sample size calculation (line 101) is unclear. Please provide further explanation or clarification.

Authors Response 6. Yes, the comment is accepted and revised and indicated by highlighted with track change as flows: To eliminate the design effect and have an adequate sample size, the sample size was doubled, and 10% of the total sample size was added to account for the non-response rate. The study was conducted in the West Shewa Zone, where 13 public health facilities were selected from 28 eligible health facilities located at the district level. People living with HIV (PLHIV) were then selected from the health facilities in each district.

7. Reviewer 1 Question 7. Figure 1: It would be more informative if the authors could display the overall number of PLHIV (People Living with HIV) in each facility and then provide the sample size for each. All the abbreviations should be added to the legend.

Authors Response 7. Yes, the comment is accepted, and the overall number of PLHIV (People Living with HIV) in each facility and the sample size for each health facility are included in Figure 1. And all the abbreviations added in the legend

8. Reviewer 1 Question 8. Patient Evaluation: It would be helpful to include information on how many patients had a positive symptom screening but did not complete further evaluation, as well as how many patients were found to have contraindications for TPT.

Authors Response 8. Yes, the comment is accepted and it was one of the limitations of this study, however contraindication for IPT was asked but all of the study participants responded that they had not reported any contraindication regarding IPT, however not included in the result.

9. Reviewer 1 Question 9. Diagnostic Methods: It is unclear why 551 patients were diagnosed with AFB smear while the initial screening was based on symptoms. Were all patients who were initially screened positive referred for AFB testing?

Authors Response 9. Thank you, the comment is accepted and corrected, indicated by highlighting with track change. It was a typographical error, corrected to 204 patients were diagnosed with AFB. Line 217, Table 2

10. Reviewer 1 Question 10. Typographical Error: There is a typographical error in the numbers; "796" is used instead of "769" in some calculations.

Authors Response 10. Thank you, the comment is accepted and corrected, indicated by highlighting with track change. It was a typographical error, corrected to 769. Table 2

11. Reviewer 1 Question 11. Table 3: I suggest restructuring Table 3. The second and third columns could be removed, and a column for p-values should be added to enhance the clarity of statistical significance.

Authors Response 11. The comment is accepted and corrected, indicated by highlighting with track change and P-Values are added

Reviewer #2

1. Reviewer 2 Question 1. The conclusion must mention the associated factors according to the title (‘Associated Factors Among People Living with HIV at Public Health Facilities of Central Ethiopia’).

Authors Response 1. The comment is accepted and corrected, indicated by highlighting with track change, and the associated factors were mentioned in the conclusion. Line 306-307

2. Reviewer 2 Question 2: This study still uses the AFB microscopy test to determine active TB. Is there no molecular rapid test available?

Authors Response 2. Yes, the comment is accepted. However, Acid-fast bacillus (AFB) tests and chest X-rays are common diagnostic tools used to detect TB infection among PLHIV at the studied health facilities (in the West Shew zone public health facilities) because molecular rapid tests were not available as indicated in the method line164-166.

3. Reviewer 2 Question 3. In data table 2, the CD4 variable (Current CD4 Count) should be added to group the CD4 count based on the diagnosis status of active or latent TB.

Authors Response 3. Yes, the comment is accepted. However, the Current CD4 count was already included in Table 2 and lines 218-120. Also, the National TB guideline in Ethiopia recommends IPT for all HIV-infected people who are unlikely to have active TB, regardless of CD4. CD4 count is not required to initiate Isoniazid Preventive Therapy (IPT) for people living with HIV (PLHIV) in Ethiopia. Similarly, Ethiopia adopted the “Universal Test and Treat” strategy to its national policy in 2016 to put all people living with HIV/AIDS (PLHIV) on antiretroviral therapy (ART) regardless of their World Health Organization (WHO) clinical stage or CD4 cell count level

4. Reviewer 2 Question 4. There were 257 (46.1%) eligible PLHIV patients who were not given IPT, please provide clear reasons for this.

Authors Response 4: Yes, the comment is accepted. However, most of these concerns were addressed in the discussion part lines 278-292, and in the result part mentioned as unknown reasons line 209. Because the study was focused only on the patients and did not address the factors related to the health facility as well as healthcare professionals, thus it could be one of the limitations of this study as further study including health facilities and healthcare professional-related factors was recommended in the conclusion part in line 321-325

---

## [Decision Letter · Decision Letter 1]

31 Jan 2025

PONE-D-24-31425R1Tuberculosis Screening, Isoniazid Preventive Therapy Coverage and Factors Associated with Active TB Diagnosis Among People Living with HIV at Public Health Facilities of Central EthiopiaPLOS ONE

Dear Dr. Aboma,

Thank you for submitting your manuscript to PLOS ONE. After careful consideration, we feel that it has merit but does not fully meet PLOS ONE’s publication criteria as it currently stands. Therefore, we invite you to submit a revised version of the manuscript that addresses the points raised during the review process.

We look forward to receiving your revised manuscript.

Kind regards,

Rebecca F. Baggaley

Academic Editor

PLOS ONE

Journal Requirements:

Reviewers' comments:

Reviewer's Responses to Questions

**Comments to the Author**

1. If the authors have adequately addressed your comments raised in a previous round of review and you feel that this manuscript is now acceptable for publication, you may indicate that here to bypass the “Comments to the Author” section, enter your conflict of interest statement in the “Confidential to Editor” section, and submit your "Accept" recommendation.

Reviewer #1: (No Response)

Reviewer #2: All comments have been addressed

2. Is the manuscript technically sound, and do the data support the conclusions?

Reviewer #1: Yes

Reviewer #2: (No Response)

3. Has the statistical analysis been performed appropriately and rigorously? 

Reviewer #1: Yes

Reviewer #2: (No Response)

4. Have the authors made all data underlying the findings in their manuscript fully available?

Reviewer #1: Yes

Reviewer #2: (No Response)

5. Is the manuscript presented in an intelligible fashion and written in standard English?

Reviewer #1: No

Reviewer #2: (No Response)

6. Review Comments to the Author

Reviewer #1: Dear Authors,

Thank you for addressing most of the comments and clarifying the questions raised in the previous review. The manuscript has improved significantly. However, there are still a few points that remain unaddressed, particularly regarding terminology and numerical inconsistencies (comment 2 and 10 from my previous review). I would like to bring up two additional comments for your consideration:

Table 3 – It would be helpful to specify whether the reported p-values correspond to the crude or adjusted OR.

Lines 255-256 – Could you clarify whether you mean that the reported TB incidence was inflated due to false-positive AFB results? Typically, AFB testing is more prone to false negatives rather than false positives, which is the greater concern with this diagnostic method.

I appreciate your efforts in revising the manuscript and look forward to your response.

Reviewer #2: (No Response)

7. PLOS authors have the option to publish the peer review history of their article (what does this mean? ). If published, this will include your full peer review and any attached files.

**Do you want your identity to be public for this peer review?** For information about this choice, including consent withdrawal, please see our Privacy Policy .

Reviewer #1: No

Reviewer #2: **Yes: ** Tutik Kusmiati

---

## [Author Response · Author response to Decision Letter 2]

1 Feb 2025

2nd Round Author's Response to Reviewers' Comments

Manuscript ID: - PONE-D-24-31425

Title of Manuscript: Tuberculosis Screening, Isoniazid Preventive Therapy, and Associated Factors Among People Living with HIV at Public Health Facilities of Central Ethiopia

Authors:

1. Mecha Aboma: abomamecha@gmail.com

2. Bayisa Abdisa

3. Gudata Imana

4. Kefalew Taye

5. Gonfa Moti

6. Merob Fufa

A Point-by-point Response to Reviewer Comments to the Authors

Reviewer #1: comments, Dear Authors,

Thank you for addressing most of the comments and clarifying the questions raised in the previous review. The manuscript has improved significantly. However, there are still a few points that remain unaddressed, particularly regarding terminology and numerical inconsistencies (comment 2 and 10 from my previous review). I would like to bring up two additional comments for your consideration: Table 3 – It would be helpful to specify whether the reported p-values correspond to the crude or adjusted OR.

Lines 255-256 – Could you clarify whether you mean that the reported TB incidence was inflated due to false-positive AFB results? Typically, AFB testing is more prone to false negatives rather than false positives, which is the greater concern with this diagnostic method.

I appreciate your efforts in revising the manuscript and look forward to your response.

Authors Response. Thank you for thoroughly reviewing my paper and helping me improve its quality. Yes, all the comments are accepted and revised and indicated by highlighting with track changes as flows.

1. Reviewer comment 1: Comments 2 and 10 from my previous review).

Authors Response 1

Yes, the comment is accepted and corrected and indicated by highlighting with track change as flow: TB infection" is replaced by "active TB throughout the document (Line 128).

Thank you, the comment is accepted and corrected, indicated by highlighting with track change. It was a typographical error, corrected to 769 (Line 25, 420).

2. Reviewer comment 2: Table 3 – It would be helpful to specify whether the reported p-values correspond to the crude or adjusted OR.

Authors Response 2

The comment is accepted and corrected, indicated in Table 3: The p-values correspond to the Adjusted OR and are specified in Table 3

3. Reviewer comment 3: Lines 255-256 – Could you clarify whether you mean that the reported TB incidence was inflated due to false-positive AFB results? Typically, AFB testing is more prone to false negatives rather than false positives, which is the greater concern with this diagnostic method.

Authors Response 3

Thank you, the comment is accepted and corrected, as indicated by highlighting with track change. In the first draft of the research paper, I submitted a sentence like this: “These differences might be attributed to false negatives reported for AFB testing and chest X-ray.” This typographical error likely occurred while I was editing and incorporating feedback from the editors and academic reviewers. I apologize for the typographical error; it has been corrected to "a false negative." Additionally, this statement pertains not only to the inflated active TB diagnoses reported in our study but also serves as a justification for the overall inconsistency of active TB diagnoses reported in our study compared to those reported by others, whether higher or lower than our findings. Thus, the comment has been genuinely accepted and corrected as flow (Line 255-259).

These differences might be attributed to false negatives reported for AFB testing and chest X-rays in diagnostic services. Additionally, inconsistent active TB diagnostics might result from inadequate diagnostic equipment, poor microscopy quality, untrained healthcare providers, substandard laboratory practices, and insufficient quality control measures, particularly in resource-limited settings.

---

## [Editor Report · Decision Letter 2]

6 Feb 2025

Tuberculosis Screening, Isoniazid Preventive Therapy Coverage and Factors Associated with Active TB Diagnosis Among People Living with HIV at Public Health Facilities of Central Ethiopia

PONE-D-24-31425R2

Dear Dr. Aboma,

We’re pleased to inform you that your manuscript has been judged scientifically suitable for publication and will be formally accepted for publication once it meets all outstanding technical requirements.

Kind regards,

Rebecca F. Baggaley

Academic Editor

PLOS ONE
---

## [Editor Report · Acceptance letter]

PONE-D-24-31425R2

PLOS ONE

Dear Dr. Aboma,

I'm pleased to inform you that your manuscript has been deemed suitable for publication in PLOS ONE. Congratulations! Your manuscript is now being handed over to our production team.

Kind regards,

on behalf of

Dr. Rebecca F. Baggaley

Academic Editor

PLOS ONE